# “Pseudo”-Secondary Treatment Failure Explained via Disease Progression and Effective Botulinum Toxin Therapy: A Pilot Simulation Study

**DOI:** 10.3390/toxins15100618

**Published:** 2023-10-18

**Authors:** Harald Hefter, Dietmar Rosenthal, Sara Samadzadeh

**Affiliations:** 1Departments of Neurology, University of Düsseldorf, Moorenstrasse 5, 40225 Düsseldorf, Germany; dietmar.rosenthal@med.uni-duesseldorf.de (D.R.); sara.samadzadeh@yahoo.com (S.S.); 2Charité–Universitätsmedizin Berlin, Corporate Member of Freie Universität Berlin and Humboldt-Unverstät zu Berlin, Experimental and Clinical Research Center, 13125 Berlin, Germany; 3Department of Regional Health Research and Molecular Medicine, University of Southern Denmark, 5230 Odense, Denmark; 4Department of Neurology, Slagelse Hospital, 4200 Slagelse, Denmark

**Keywords:** simulation, duration of botulinum toxin action, secondary treatment failure, neutralizing antibodies, focal dystonia, long-term outcome of botulinum neurotoxin therapy

## Abstract

Background: The objective of this study was to provide evidence from a simple simulation. In patients with focal dystonia, an initial good response to botulinum neurotoxin (BoNT) injections followed by a secondary worsening does not necessarily arise from an antibody-induced secondary treatment failure (NAB-STF), but may stem from a “pseudo”-secondary treatment failure (PSEUDO-STF). Methods: The simulation of the outcome after BoNT long-term treatment was performed in four steps: 1. The effect of the first single BoNT injection (SI curve) was displayed as a 12-point graph, corresponding to the mean improvement from weeks 1 to 12. 2. The remaining severity of the dystonia during the nth injection cycle was calculated by subtracting the SI curve (weighted by the outcome after n − 1 cycles) from the outcome after week 12 of the (n − 1)th cycle. 3. A graph was chosen (the PRO curve), which represents the progression of the severity of the underlying disease during BoNT therapy. 4. The interaction between the outcome during the nth BoNT cycle and the PRO curve was determined. Results: When the long-term outcome after n cycles of BoNT injections (applied every 3 months) was simulated as an interactive process, subtracting the effect of the first cycle (weighted by the outcome after n − 1 cycles) and adding the progression of the disease, an initial good improvement followed by secondary worsening results. This long-term outcome depends on the steepness of the progression and the duration of action of the first injection cycle. We termed this response behavior a “pseudo”-secondary treatment failure, as it can be compensated via a dose increase. Conclusion: A secondary worsening following an initial good response in BoNT therapy of focal dystonia might not necessarily indicate neutralizing antibody induction but could stem from a “PSEUDO”-STF (a combination of good response behavior and progression of the underlying disease). Thus, an adequate dose adaptation must be conducted before diagnosing a secondary treatment failure in the strict sense.

## 1. Introduction

Botulinum neurotoxins (BoNTs) are single-strain clostridial products [1], which have to be cleaved to become the most toxic proteins [2,3,4,5]. For clinical applications, several BoNT/A preparations, which have to be diluted with saline, and a ready-to-use BoNT/B preparation are available and licensed [6,7,8,9]. The BoNT/A preparations differ in their contents of activated BoNT/A (for details, see the prescribing information published by the authors of [6,7,8,10]). They also differ in their contents of inactivated neurotoxin, botulinum toxin fragments, or complex proteins, which do not enhance biological functions but may serve as adjuvants [11] and increase the antigenicity of a BoNT preparation [12,13,14]. BoNT/A and BoNT/B dilutions have to be injected to reduce muscle end-plate activity, glandular hypersecretion, or transmitter release [15]. This traumatic mode of application may lead to the stimulation of dendritic cells [16] and the recognition of components of BoNT by T-cells [17,18]. Therefore, the detection of BoNT by the immune system can hardly be avoided when long-term treatments of BoNT are performed with many repeated injections [18]. There is a clinically relevant risk (depending on a variety of other factors, such as dose per session, duration of treatment, or the number and location of injection sites) that neutralizing antibodies (NABs) [19,20] are induced, which reduce the biological function of a BoNT preparation and cause an antibody-induced secondary treatment failure (NAB-STF) [21,22,23,24,25].

Estimation of the risk to develop an antibody-induced STF is difficult and a matter of debate due to various reasons [22,23,24]. On the one hand, different clinical, as well as laboratory tests (to detect NABs), are used, which clearly differ in terms of their sensitivities [26,27]. On the other hand, there is no formal criterion for a STF, except that a secondary worsening occurs after an initial good response. It has been repeatedly reported that at least 50% of patients with STF do not have NABs [28,29], and some authors have come to the conclusion that the discussion on NAB-induced STF is “much ado about nothing” [28]. It has been argued that in a fairly large percentage of patients with suspected STF, the use of new guidance techniques for BoNT injections, the choice of appropriate muscles, and the increase in dose clinical outcome can all be improved [22,23]. Others have argued that these cases of STF have not developed an STF in the strict sense but have been treated insufficiently. We, therefore, prefer the term “pseudo”-secondary treatment failure (PSEUDO-STF) to describe this situation of insufficient therapy [30].

The question is why this situation of insufficient therapy may develop, although previous BoNT injections have been more successful. Of course, NAB induction and the subsequent continuous increase in NAB titers with the declining duration of BoNT action provide an explanation as long as the dose per session is kept constant [21]. But even when NABs are present, an increase in the dose may compensate for the decline in the duration of BoNT action so that a partial STF may persist for years [21].

Another equally likely explanation for secondary worsening after an initial good response is provided by the progression of the underlying disease. The difficulty to diagnose a progression of a BoNT-treated disease, however, is that with the onset of BoNT therapy, the natural course of disease severity cannot be observed anymore. In principle, it is known from the pre-BoNT era that focal dystonia (especially CD) usually becomes worse with time [31]. But in an individual patient, it may be impossible to detect a worsened effect, especially in the beginning of the BoNT therapy, due to the clear-cut improvement induced via the BoNT therapy. BoNT injections in patients with CD, with a sufficiently high dose per session, will lead to a clinical improvement between 20% and 60% [32,33,34] in most of the treated patients (for an overview, see Table 8.2 published by the authors of [35]). This obscures a slow progression of the underlying disease.

However, there are clear clinical hints (at least for some BoNT-treated disease entities) that a progression may occur despite a clinical improvement under BoNT. When long-term BoNT-treated patients with CD are asked to remember which symptoms they had at the onset of BoNT therapy, and to compare that spectrum of symptoms with the spectrum of symptoms they now have after years of BoNT therapy, they will report that some symptoms have been improved, but that other symptoms have developed during the course of BoNT therapy. Thus, they report a progression of CD despite having a satisfactory improvement of some symptoms of CD [36]. Furthermore, the spread of symptoms (e.g., from the neck muscles to the shoulder muscles or the proximal arm muscles) occurs in at least 30% of BoNT-treated patients with CD. This speaks for a progression of CD during BoNT therapy [37,38], especially in the central nervous system (CNS). In this situation, a progression of CD may initially be compensated by the application of the usual dose of BoNT, but a time may come where the usual dose of BoNT cannot fully compensate the worsening of the underlying disease anymore. As a result, a secondary worsening will occur, which may be confused with an NAB-STF, especially when the dose or the injection scheme is not adapted. We think that many BoNT centers have indeed adapted the dose or the injection scheme to the progression of the underlying disease. This is a likely explanation for why studies on long-term BoNT treatment often report dose increases with increasing treatment duration [39,40].

We performed a simulation study to demonstrate that a PSEUDO-STF can easily be explained by a combination of a clinical improvement of BoNT and a progression of the underlying disease. With that, we then provided an explanation for why, in a fairly large percentage of STF patients, no NABs were detected.

## 2. Results

### 2.1. Examples of Self-Assessment of the First BoNT Injection

A daily assessment of the remaining severity of this disease by a patient (p) as a percentage of the initial disease’s severity during the first injection cycle (post-initiation of BoNT therapy) revealed a distinctive curve (RS_1,p_). Following a few days, a progressive improvement was noted, peaking between 3 and 5 weeks. Subsequently, the clinical effect of the BoNT injection began to wane, and the RS_1,p_ curve started to rise. By around day 90, when the next injection was scheduled, a minor residual improvement (q_p_) relative to the baseline was observed. This modest improvement differed among patients, was consistent within a patient, and was pivotal for the current simulation.

For illustration, two examples are showcased in Figure 1. One patient (p1; represented using full circles in Figure 1A) recorded a stellar response, with a residual improvement of 20% (q_p1_ = 0.2). In contrast, another patient (p2; denoted using open circles in Figure 1B) witnessed a less pronounced peak effect, documenting a residual improvement of merely 10% (q_p2_ = 0.1). To underscore the impact of the q-value, results utilizing a q-value of 0.2 were displayed on the left side of all figures, while results using a q-value of 0.1 were presented on the right side.

### 2.2. Examples of Self-Assessment of Four Consecutive BoNT Injections

A self-assessment by a patient (denoted as p), detailing the residual severity of their disease across a number (n) of consecutive BoNT injections, administered every 3 months, produces a progressively decreasing severity profile, depicted as an RS_n,p_ curve. As each subsequent injection commences from a diminished disease severity baseline, the residual severity lessens with each injection. Although the effect of an individual injection declines, the cumulative improvement increases. This pattern is illustrated in Figure 2 with two distinct patients. Patient p3, represented by full circles, exhibited an outstanding response, with his RS_n,p3_ curve nearing the 60% improvement mark just after four injections. Conversely, patient p4, denoted by open circles, showed a more moderate improvement during their initial injection cycle. Consequently, their staircase-patterned RS_n,p4_ curve descended at a leisurely pace, achieving an improvement range between 35% and 40% following four injections (see Figure 2B).

### 2.3. Simulation of the Clinical Effect of Consecutive BoNT Injections

The residual severity of a patient’s disease (RS_n,p_), following n successive BoNT injections, was derived from the outcome of the first injection cycle. SI_1,p_(wi) denoted the average improvement during week wi, where 1 ≤ i ≤ 12. SI_1,p_(w1) was 0. The residual effect after week 12, SI_1,p_(w12), was termed q. The residual severity of the disease for patient p during the first cycle was RS_1,p_(wi) at week wi (1 ≤ i ≤ 12) and was equivalent to 1 − SI_1,p_(wi).

For the mth cycle (m > 1), the residual severity RS_m,p_(wi) at week wi was calculated as:RS_m,p_(wi) = RS_m−1,p_(w12) − RS_m−1,p_(w12)*SI_1,p_(wi), 1 ≤ i ≤ 12.(1)

As SI_1,p_(w1) was 0, RS_m,p_(w1) is equivalent to RS_m−1,p_(w12). Hence, RS_m,p_(w12) = RS_m−1,p_(w12) − RS_m−1,p_(w12)*q = (1 − q)*RS_m−1,p(_w12).

Given that RS_m−1,p_(w12) equals (1 − q)^m−1^, RS_m,p_(w12) yielded a value of (1 − q)^m^. This facilitates the easy computation of the residual severity post the mth injection cycle.

In Figure 3A (depicted by full circles), a single injection curve, SI_1,p_(w), is shown, displaying an exceptional response, q_p1_ = 0.2. In Figure 3B (shown by open circles), SI_1,p_(w) presents a moderate response, q_p2_ = 0.1, to the initial BoNT injection. The residual severity curve, RS_m,p_(w), was determined for 20 cycles (full circles), corresponding to a treatment span of 240 weeks (or 4.6 years). Both RS_m,p_(w) curves showed a decreasing trend, reflecting an amplified improvement. However, the impact of a singular injection progressively diminishes with successive BoNT injections (as seen in Figure 3A,B). After ten injection cycles, the improvement neared 90% for the excellent response curve, q_p1_ = 0.2 (Figure 3A), and about 65% for the moderate response curve, q_p2_ = 0.1 (Figure 3B).

### 2.4. Simulation of Disease Progression and Effective BoNT Treatment

In clinical practice, after the onset of BoNT therapy, the natural progression of the underlying disease became unobservable due to the improvement initiated by the BoNT injections. However, in these simulations, the progression of the disease, denoted by PRO_m,p_(w), can be seamlessly incorporated into the RS_m,p_(w) curve. Figure 4A,B depict the trajectory of the remaining severity when a linear function, representing a 60% progression of the underlying disease over 240 weeks (or 4.6 years), is incorporated into the RS_m,p_(w) curves shown in Figure 3A,B.

This linear function was selected based on clinical observations, where patients with CD typically have an initial TSUI score of around 10. Exceptional cases exhibit a TSUI score [41] ranging between 14 and 16. The following linear function was added to RS_m,p_(wi):PRO_m,p_(wi) = 1 + 0.6*(m*wi/240) with 1 ≤ m ≤ 20 and 1 ≤ wi ≤ 12.(2)

Summation of the RS_m,p_ and the PRO_m,p_ curves resulted in the total clinical effect curve termed the CLEF_m,p_ curve. Both resultant curves, CLEF_m,p1_(wi) = RS_m,p1_(wi) + PRO_m,p1_(wi) (seen in Figure 4A) and CLEF_m,p2_(wi) = RS_m,p2_(wi) + PRO_m,p2_(wi) (seen in Figure 4B), depicted an initial improvement followed by subsequent deterioration, appearing as though a secondary treatment failure emerged. Notably, the SI_1,p1_(w) curve, which was used in simulating RS_m,p1_(wi) in Figure 4A, and the SI_1,p2_(wi) curve, which was employed for RS_m,p2_(wi) in Figure 4B, remained consistent across all 20 treatment cycles wi.

To showcase the impact of the progression of the underlying disease, another linear function was integrated into the RS_m,p_(wi) curves from Figure 3A,B. This linear function mimics a milder disease progression at 40%:PRO_m,p_(wi) = 1 + 0.4*(m*wi/240) with 1 ≤ m ≤ 20 and 1 ≤ wi ≤ 12.(3)

In this scenario, a pronounced initial improvement followed by secondary deterioration was only evident when the SI_1,p1_(wi) curve, which has an exceptional response q_p1_ = 0.2 (seen in Figure 4C), was used for simulating the RS_m,p1_(wi) curve and the overall clinical outcome CLEF_m,p1_(wi). For the CLEF_m,p2_(wi) curve, employing the SI_1,p2_(wi) curve with a moderate response q_p2_ = 0.1 (seen in Figure 4D), there was an initial improvement observed but without a distinct secondary decline. This is because the diminished effect of progression was offset by the ongoing improvement. This simulation suggests that over the years, the secondary deterioration due to the progression of the underlying disease may be entirely masked by the BoNT-induced enhancement, particularly in patients experiencing a low-to-moderate impact from BoNT therapy.

## 3. Discussion

### 3.1. General Aspects of the Choice of Method for the Mathematical Modeling of Therapeutic Outcomes

Simulations of BoNT activity have predominantly been concentrated on the biomechanical alterations in intramuscular force generation and the adaptation of neuromuscular reflex circuits, utilizing advanced models and simulation techniques [41]. Contrarily, our current study addressed a broader perspective of BoNT therapy, employing straightforward arithmetic methods familiar to every clinician.

Our choice to employ the response to the initial BoNT injection (referred to as the SI curve) as a foundation for simulating BoNT therapy stems from examining numerous patients’ self-assessment curves. Notably, the diminished response observed in successive injections, compared to the first, prompted us to employ an iterative approach. Here, each new cycle begins from a heightened state of improvement but anticipates a subsequently reduced reaction to the forthcoming injection. Interestingly, when queried about the trajectory of disease severity preceding BoNT therapy, fewer than half of the patients illustrated a linear progression of CD [30]. Despite this, our initial simulation strategy embraced a linear function to represent the progression of CD. It is worth noting that non-linear disease progression models are equally feasible, but, regrettably, information remains scant regarding disease progression amidst BoNT therapy, especially concerning CD.

This simulation study juxtaposed two counteracting processes: the sustained improvement achieved via recurrent BoNT injections, and the relentless progression of the inherent disease. The significance of our simulation lies in its versatility; it can accommodate varying q-values (which correspond to different efficacy durations) and diverse progression rates (as exemplified in Figure 4). Such flexibility facilitates a clear demonstration of how these parameters interplay and establishes which one exerts a more pronounced influence throughout the treatment’s span.

### 3.2. Increasing Improvement with Repetitive Injections of BoNT

For most indications in clinical practice, repeated injections are administered. Typically, a subsequent BoNT injection is administered before the effect of the prior one has completely waned. This approach, which results in a stepwise enhancement, is more practical than administering a single BoNT injection, then delaying the next dose until symptoms nearly return to baseline.

In our institution, the vast majority of patients (over 85%) receive injections every 12 to 13 weeks. Through adopting this regimen, we have observed a consistent improvement ranging from 50% to 60%, as validated by both physician evaluations and patients’ self-assessments at the conclusion of an injection cycle [33,42]. However, due to the stringent four-week lockdown imposed during the spring of 2021 amidst the COVID-19 pandemic, patients faced extended intervals between injections, documenting a daily decline of around 1% during this lockdown. Those who waited beyond four weeks noted a deterioration exceeding 30%, significantly diminishing their quality of life [43].

However, this approach of beginning the next injection cycle well before the effect of the previous injection has fully subsided makes it challenging to determine the duration of the effect of a single injection. In this simulation, we assessed the improvement on the final day of an injection cycle compared to the disease’s severity level on the day the injection was administered. This assessment was set in place of measuring the actual duration of a single injection’s clinical effect (see Figure 1).

We recently discussed a case involving a patient with complex regional pain syndrome and dystonia in his left hand. This patient responded exceptionally well to high doses of incobotulinum neurotoxin. He received a subsequent injection even before the effects of the initial injection began to wear off. Notably, his improvement was 60% after the first injection, and this was observed before the second injection administered 100 days after the commencement of incoBoNT/A therapy [42]. After four injections, this patient showed a remarkable 95% improvement. This aligns closely with our simulation prediction of improvement, calculated as 1 − (1 − 0.6)^4^ = 0.974 or 97%. It emphasizes the importance of evaluating the severity of symptoms at the conclusion of an injection cycle, right before administering the next dose.

### 3.3. Progression of the Disease despite Improvement during BoNT Therapy

The long-term outcomes of deep brain stimulation (DBS) operations for patients with primary dystonia are influenced by the disease’s duration prior to the procedure. Specifically, the longer the symptoms have been present, the more challenging it becomes to achieve improvement [44,45]. A parallel observation has been noted for BoNT injection therapy in patients with cervical dystonia (CD) [36].

One potential explanation is that while CD originates in the CNS, BoNT injection therapy targets the neuromuscular periphery as a symptomatic treatment. While BoNT injections effectively alleviate a few focal dystonia symptoms, a progression from a focal to a multifocal status has been observed in up to 30% of CD patients [36,37,38]. Still, most long-term BoNT-treated CD patients do not exhibit signs of secondary treatment failure.

There is no conclusive evidence available suggesting that BoNT therapy reduces or influences the progression of the underlying disease. In this study, we simulated the interplay between BoNT therapy effects and disease progression by adding a linear function, PRO_m,p_(w), to the simulated remaining severity curve, RS_m,p_(w). While this interaction may not be linear, our initial approach offers predictions aligned with clinical observations. Nonetheless, further studies on disease severity in patients not undergoing BoNT therapy would be beneficial to better understand the natural progression of focal dystonia.

Measurements of disease progression significantly depend on the sensitivity of the scoring system being employed. Our insights were grounded in patients’ self-assessments of symptom severity using a 21-point Likert scale and the TSUI score [46] for those with CD. These scores were taken right before a CD patient received their injection. Additionally, we leveraged the method where patients chart their disease course, which assists in identifying those with a secondary worsening effect after an initially positive response (for further details, see the study published by the authors of [30]).

### 3.4. PSEUDO-STF Results from a Good Clinical Response to BoNT and Disease Progression

In this study, we termed the secondary worsening, which arises from the interplay between an effective BoNT treatment and the progression of the underlying disease, as “PSEUDO-STF”. This distinction helped in differentiating it from an NAB-induced STF and the interactions between BoNT therapy and disease progression. In patients who have been treated for an extended period with a consistent dose, slow disease progression may go unnoticed for years (refer to Figure 4). Eventually, however, a secondary worsening is inevitable. When these patients are tested for NABs, their results typically return negative.

With appropriate dose adjustments, this “PSEUDO-STF” can be largely compensated for, a practice that is likely adopted in many centers. Numerous long-term outcome studies have shown that BoNT doses tend to increase over the treatment’s duration [33,39,40]. As evidenced by simulations in Figure 3A,B, when consistent BoNT doses are administered every three months, and there is no progression of the underlying disease, continuous improvement can be anticipated. Yet, without dose adjustments, and with the onset of mild disease progression (even with symptom improvement), a “PSEUDO-STF” will manifest, as shown in Figure 4. We believe that many centers increase the dose during prolonged BoNT therapy to prevent secondary worsening stemming from disease progression, even if this progression has not been explicitly diagnosed. To better comprehend and regulate this dose adaptation, a multicenter follow-up study is advisable.

## 4. Conclusions

The present study, simulating repeated injections every 3 months with different durations of efficacy to a BoNT injection accompanied with a slowly linear progression of the underlying disease, has clearly demonstrated that a slowly progressive worsening after an initial good response may result from the interaction between a good response to BoNT therapy and the disease progression, and not necessarily from a NAB-induced STF. Therefore, a NAB-induced-STF should only be diagnosed in a long-term continuously BoNT-treated patient after the attempt has failed to enhance clinical improvement via dose adaptation (or via an increase to prolong the duration of the efficacy of a single injection) and adjustment of the injection scheme. Otherwise, such a patient is inadequately treated and suffers from a “PSEUDO-STF”. A “PSEUDO-STF” is difficult to distinguish from an NAB-induced-STF only on the basis of clinical outcome measures.

## 5. Strengths and Limitations of the Simulation Study

The current study simulated repeated injections every 3 months, with varying durations of efficacy and a slow linear progression of the underlying disease. This suggests that a gradual worsening after an initial positive response can result from the interaction of a successful BoNT therapy and the disease’s progression, rather than necessarily being due to an NAB-induced STF. Therefore, a diagnosis of an NAB-induced STF should only be considered in patients under continuous, long-term BoNT treatment if efforts to improve clinical outcomes through dose adjustments (specifically, increases meant to prolong the efficacy of a single injection) and changes to the injection approach are unsuccessful. If this is not addressed correctly, a patient might be mistreated, resulting in a “PSEUDO-STF”. Moreover, it was challenging to differentiate a “PSEUDO-STF” from an NAB-induced-STF based purely on clinical results.

## 6. Materials and Methods

### 6.1. Self-Assessment Curves

In the BoNT outpatient department of the Neurology Clinic in Düsseldorf, the assessment of the efficacy of BoNT therapy by the patients was standardized by training the patients to assess the effect of a BoNT injection by means of a 21-point Lickert scale ranging from 0%, 5%, 10%, …to 90%, 95%, and 100%. Patients scored their actual severity as a % of the severity of their disease at the onset of BoNT therapy. In Figure 1A,B, the self-assessment curves of two typical patients (p1 and p2) suffering from a cervical dystonia are presented. In Figure 2A,B, the self-assessment curves of two further patients (p3 and p4) of the effect of the first 4 injection cycles of their CD are presented.

### 6.2. Simulation

This simple simulation of the effect of repetitive single injections of BoNT was performed in four steps:
Twelve data pairs were selected which yielded a simple (x,y) curve, (SI_1,p_(w)), representing the 12 mean values of remaining severity during the first BoNT injection cycle for week w1, w2, 2026, w12 for a patient, p. Two examples are presented in Figure 1A,B. SI_1,p_(w1) = 0, since the improvement usually began at the end of the first week after BoNT injection. The value 1-SI_1,p_(w12) was called q_p_, and represents the residual improvement after the first injection cycle just before the next injection is performed. In Figure 1A, a SI_1,p1_(w) curve is presented, with q_p1_ = 0.2, and in Figure 1B, a SI_1,p2_(w) curve is displayed, with q_p2_ = 0.1.The simulated remaining severity curve resulted from an iteration process; for the calculation of the remaining disease severity, RS_m,p_(w), during the mth cycle and week wi, the value RS_m−1,p_(12) and the SI_1,p_(w) curve were used, which yielded the formula (I):RS_m,p_(wi) = RS_m−1,p_(w12) m− RS_m−1,p_(w12)*SI_1,p_(wi), with RS_0,p_(w12) = 1 and 1 ≤ i ≤ 12. This implies that RS_1,p_(w1) = 1, since SI_1,p_(w1) = 0, and that RS_1,p_(w12) = 1 − q. The value of RS_m,p_(w12) was (1 − q)^m^ (see the Results section). Therefore, the remaining severity, RS_m,p_(w12), after m injections can be easily calculated and only depends on q (see Figure 3A,B).Then, a linear function (PRO_m,p_(w) curve) was selected, which characterizes the progression of the underlying disease for 240 weeks (=20 BoNT injection cycles = 4.6 years): PRO_m,p_(wi) = 1 + WF_p_*(m*wi/240), with 1 ≤ m ≤ 20 and 1 ≤ i ≤ 12). WF_p_ is a weighting factor which depends on the patient p.


This implies that the disease severity progresses from 1 to 1 + WF_p_ at the end of the 20th cycle. In Figure 4, two different examples are chosen for this linear function; in Figure 4A,B, the weighting factor WF_p_ is equal to 0.6, and in Figure 4C,D, WF_p_ = 0.4.
D.Finally the simulation of the overall clinical effect (CLEF_m,p_(w) curve) results from the sum of the remaining severity, RS_m,p_(w), and the progression of the disease, PROm,p(w):CLEF_m,p_(wi) = RS_m,p_(wi) + PRO_m,p_(wi) with 1 ≤ m ≤ 20 and w1 ≤ wi ≤ w12.(4)

In Figure 4A,B, CLEF_m,p_(w) curves are presented, with WF_p1_ = 0.6, whereas in Figure 4C,D, WF_p2_ = 0.4 is used.

### 6.3. Statistics

In the present study, no statistical procedures were applied. The simulation was performed using the program EXCEL^®^, which is part of the Microsoft Office^®^ package.

## Figures and Tables

**Figure 1 toxins-15-00618-f001:**
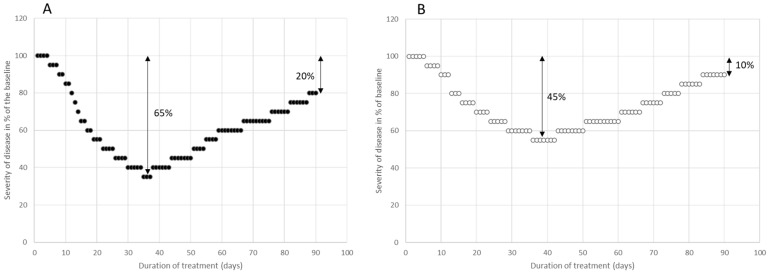
Daily self-assessment of disease severity using a 21-point Likert scale throughout the initial BoNT injection cycle in two CD patients. The severity of CD at the beginning of BoNT therapy was established at 100%. Patient p1 demonstrated an outstanding response, peaking at an improvement of 65%, and showing a 20% residual improvement after 3 months (**A**). Conversely, patient p2 had a positive response to the first injection, but with a more modest peak improvement of 45%, along with a 10% residual improvement (**B**).

**Figure 2 toxins-15-00618-f002:**
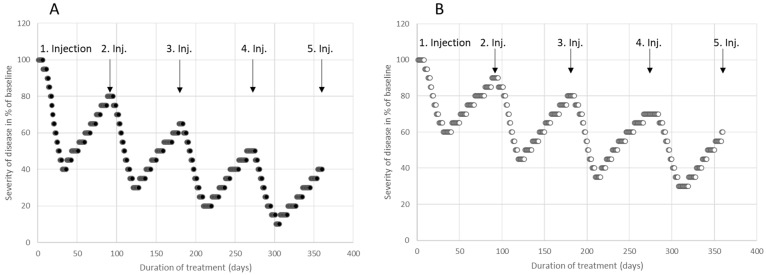
Daily self-assessment of the disease severity (using a 21-point Lickert scale) during the first 4 BoNT injection cycles of two further patients with CD. The severity of CD at the onset of BoNT therapy was set to 100%. Patient p3 (**A**) experienced an excellent response, with a peak effect of 60% during the first injection cycle, a 20% residual improvement after 3 months, and a residual improvement of 60% after 4 injection cycles. Patient p4 (**B**) also responded well to the first injection, with a peak effect of 40% and a residual improvement of 10%. After 4 injections, patient p4 had a residual improvement of 40%.

**Figure 3 toxins-15-00618-f003:**
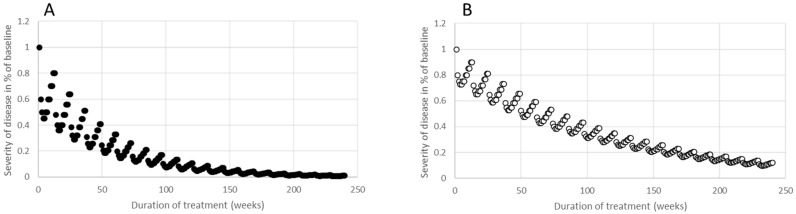
In part (**A**) a SI_1,p1_(w) curve, which showcases a peak effect of 55% and a residual improvement of 20%, was used for the simulation of 20 BoNT injections administered every 12 weeks. The residual severity of the disease, RS_m,p1_(w), was depicted based on the SI_1,p1_(w) curve. In part (**B**), a SI_1,p2_(w) curve, with a peak effect of 27,5% and a residual improvement of 10%, was employed for the simulation of 20 BoNT injections given at 12-week intervals. The residual severity of the disease, RS_m,p2_(w), was illustrated using the SI_1,p2_(w) curve.

**Figure 4 toxins-15-00618-f004:**
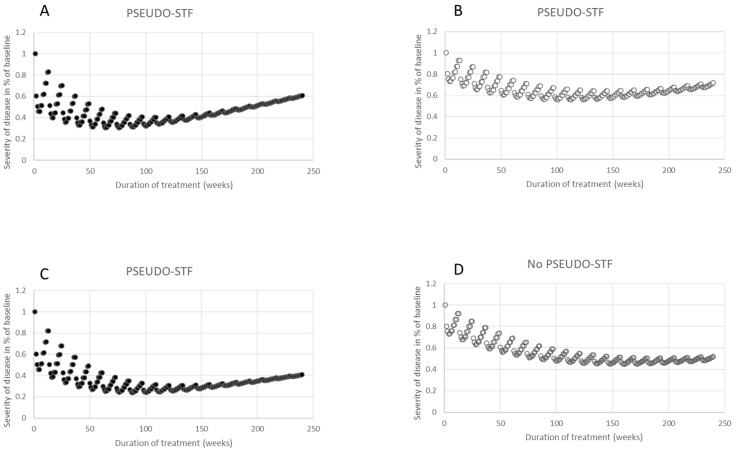
In (**A**,**B**), a linear function illustrating a progression of 60% over 4.6 years was applied to the RS_m,p_(w) curves presented in Figure 3A,B. In (**C**,**D**), a linear function representing a 40% progression across the same 4.6-year duration was incorporated into the RS_m,p_(w) curves from Figure 3A,B. Notably, part D did not exhibit any secondary worsening, despite all computations being carried out in a manner consistent with (**A**–**C**).

## Data Availability

Data are available upon request related to the restrictions of privacy or ethics. The data presented in this study are available upon request from the corresponding author.

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
