# Peer review of "“Pseudo”-Secondary Treatment Failure Explained via Disease Progression and Effective Botulinum Toxin Therapy: A Pilot Simulation Study"

_toxins, 2023, doi:10.3390/toxins15100618_

Round 1

Reviewer 1 Report

The authors of the manuscript address the very interesting topic of secondary treatment failure of botulinum toxin therapy. The question of whether neutralising antibodies or disease progression are responsible for this secondary therapy failure is addressed. The question is asked and how it can be explained that in a large number of patients in whom secondary therapy failure occurs, no neutralising antibodies are found. They present a mathematical model that simulates the clinical effect of repeated treatment with botulinum toxin and disease progression. From this model, the authors deduce that secondary treatment failure can be caused by disease progression.
This approach is interesting and illustrates the clinical question.

I have questions and comments about the implementation and interpretation.

The mathematical approach assumes strongly idealised conditions. The patients' statements on the effects of a treatment rarely correspond to the two examples presented, but are much more varied. The information that patients give about a treatment effect is also influenced by other factors, such as pain, depression, concomitant diseases, so in my routine the indication of the clinical effect is highly variable within but also between treatment cycles. Another assumption, that there is a better and clearer effect from one treatment to the next, also appears to be highly idealised and not very representative of reality in this form. Furthermore, a disease progression rate of 60% is assumed. This also seems to me to be a value that cannot be generally assumed. The assumption of a linear increase in disease progression also seems very simplified and not sufficiently supported by data. From studies of the Dystonia Coalition, a spread and worsening of the dystonia is only described in some patients, while many patients remain stable in terms of the disease expression after an initial development of the symptoms.

My basic criticism of the mathematical approach is that, with the strong simplification, the model represents two effects running against each other on a course of treatment. However, this is so obvious that it is not clear to me why a mathematical approach is necessary for this.

Author Response

The authors of the manuscript address the very interesting topic of secondary treatment failure of botulinum toxin therapy. The question of whether neutralising antibodies or disease progression are responsible for this secondary therapy failure is addressed. The question is asked and how it can be explained that in a large number of patients in whom secondary therapy failure occurs, no neutralising antibodies are found. They present a mathematical model that simulates the clinical effect of repeated treatment with botulinum toxin and disease progression. From this model, the authors deduce that secondary treatment failure can be caused by disease progression.
This approach is interesting and illustrates the clinical question.

I have questions and comments about the implementation and interpretation.

The mathematical approach assumes strongly idealised conditions. The patients' statements on the effects of a treatment rarely correspond to the two examples presented, but are much more varied. The information that patients give about a treatment effect is also influenced by other factors, such as pain, depression, concomitant diseases, so in my routine the indication of the clinical effect is highly variable within but also between treatment cycles. Another assumption, that there is a better and clearer effect from one treatment to the next, also appears to be highly idealised and not very representative of reality in this form. Furthermore, a disease progression rate of 60% is assumed. This also seems to me to be a value that cannot be generally assumed. The assumption of a linear increase in disease progression also seems very simplified and not sufficiently supported by data. From studies of the Dystonia Coalition, a spread and worsening of the dystonia is only described in some patients, while many patients remain stable in terms of the disease expression after an initial development of the symptoms.

My basic criticism of the mathematical approach is that, with the strong simplification, the model represents two effects running against each other on a course of treatment. However, this is so obvious that it is not clear to me why a mathematical approach is necessary for this.

Reviewer 1 nicely summarizes our paper.

Reviewer 1 is right: patients´ statements usually describe the complains of the patients without mentioning numbers. However, this situation changes when patients are asked to assess the treatment effect using scales. We now mention that a Lickert scale is used to standardize patients´ assessment of the treatment effect.

Since our patients usually assess their severity by means of a 21point Lickert scale we have many examples (in different disease entities) demonstrating the staircase-like improvement injection by injection from onset of BoNT therapy.

We have used a progression rate of 60% (Figs. 1-4 left side) and of 40% (Fig. 1-4 right side) to demonstrate the influence of the progression rate. This is now mentioned more clearly.

The reason to apply idealizing mathematical procedures is to detect a principle which is buried under the natural variability (as e.g. calculation of a linear regression line and determination of the correlation coefficient).

We are in line with reviewer 1 that the majority of patients (about 2/3) remain stable and nether present hints for a STF. We emphasize that in the revised manuscript in the discussion.

Our answer to this basic criticism is straight forward: When two effects run against each other it is totally unclear whether one effect is stronger than the other. Our simulation shows that the BoNT effect dominates in the beginning and that disease progression may become relevant after 20 injections. This is now mentioned explicitly at the beginning of the discussion.   

Reviewer 2 Report

Treatment failure after botulinum toxin therapy is an important issue. The main problem with this paper is that it is difficult to understand. Partly this is because of the over-use of non-obvious abbreviations (eg SI-curve is a confusing term). The confusion begins with the Abstract, line 11 (...weeks 1,2,...12.2) - what does this refer to? The introduction, in discussing that  immune response to BoNT does not mention that this might be dependent on site of injection, or amount of toxin injected. Part of the authors' argument is that during BoNT treatment it is not possible to observe disease progression, but they do not appear to cite information prior to the use of BoNT. The results are presented in a confusing way. For example the legend to figure 1 does not enable the figure to be understood, as it is incomplete without reading the text. It would be better presented with the data for both patients plotted on a single graph. Figure 2 is more confusing, where part B (not referred to in the text or legend) is unclear. Their basic conclusion is clear, but understanding the maths is made more difficult by their abbreviations. Understanding line 26 requires some thought - it is not well presented. Figure 3 needs to be explained better, and the legend needs to discuss parts C & D.

For this paper to be of interest to a wide readership, it needs to be explained much more clearly.

The English is generally good, but could definitely be improved with a little help from a native speaker.

Author Response

Treatment failure after botulinum toxin therapy is an important issue. The main problem with this paper is that it is difficult to understand. Partly this is because of the over-use of non-obvious abbreviations (eg SI-curve is a confusing term).

The confusion begins with the Abstract, line 11 (...weeks 1,2,...12.2) - what does this refer to?

The introduction, in discussing that  immune response to BoNT does not mention that this might be dependent on site of injection, or amount of toxin injected.

Part of the authors' argument is that during BoNT treatment it is not possible to observe disease progression, but they do not appear to cite information prior to the use of BoNT.

The results are presented in a confusing way.

For example the legend to figure 1 does not enable the figure to be understood, as it is incomplete without reading the text.

It would be better presented with the data for both patients plotted on a single graph.

Figure 2 is more confusing, where part B (not referred to in the text or legend) is unclear.

Their basic conclusion is clear, but understanding the maths is made more difficult by their abbreviations.

Understanding line 26 requires some thought - it is not well presented.

Figure 3 needs to be explained better, and the legend needs to discuss parts C & D.

For this paper to be of interest to a wide readership, it needs to be explained much more clearly.

Comments on the Quality of English LanguageThe English is generally good, but could definitely be improved with a little help from a native speaker.

For the help of the reader, we have added a list of abbreviations at the end of the Methods section.

The term SI-curve was explained by the preceding words: “the first single BoNT injection (SI-curve)”.

In line 10 four steps were mentioned: 1) to 4). The confusing 12.2) results from the fact that before step 2) begins weeks 1 to 12 are mentioned. This is made clear now.

We now have added several factors that may influence the immune response to BoNT.

Of course, we have checked the literature of the pre-BoNT-era. We are thankful for the reminder to add some of these references.

We have now extended the legend to Fig. 1.

To avoid confusion and remain consistent we have presented results with q=0.2 on left side and with q=0.1 on the right side of all figures (Fig. 1 to 4).

The legend to Fig. 2 is also extended now.

It is impossible to apply mathematical procedures without use of abbreviations. We now explain the abbreviations in more detail.

Obviously, reviewer 2 does not mean line 26 which contains key words. We think he means the Conclusions section following line 260. We have extended this short paragraph now.

Reviewer 2 is absolutely right. Unfortunately, we have up-loaded a previous version of Fig. 3. The actual version of Fig. 3 only contents part A and B and is consistent with the main text and legend to the figure.

We have substantially revised the manuscript.

Reviewer 3 Report

Strengths of the Article:

  1. Novel Approach: The article employs a simulation-based approach to investigate the complex interplay between the effects of Botulinum neurotoxin (BoNT) therapy and disease progression. This approach provides a fresh perspective on understanding long-term treatment outcomes.
  2. Clinical Relevance: The study addresses an important clinical concern – the occurrence of secondary treatment failures in patients undergoing BoNT therapy. By modeling how the response to treatment and disease progression interact, the study sheds light on potential reasons for variations in treatment outcomes.
  3. Patient Self-Assessment: The use of patient self-assessment, including a 21-point Lickert-scale, adds credibility to the study's findings. Patient-reported outcomes are a valuable source of information for assessing treatment efficacy.
  4. Real-world Application: The study acknowledges the clinical practice of starting the next BoNT injection cycle before the full effect of the previous injection has diminished. This real-world context makes the simulation results more applicable to clinical decision-making.

Limitations of the Article:

  1. Simplified Model: The simulation employs a linear model to represent disease progression and BoNT therapy effects. While this simplification facilitates mathematical analysis, real-world disease progression is likely more complex and nonlinear.
  2. Lack of Validation: The simulation results are not validated against actual patient data or clinical trials. While the approach is insightful, validation against real-world outcomes would strengthen the conclusions.
  3. Single Disease Focus: The study primarily focuses on cervical dystonia, limiting the generalizability of its findings to other disease entities treated with BoNT.
  4. Lack of Statistical Analysis: The article does not utilize statistical analysis to support its conclusions, potentially leaving room for interpretation bias.
  5. Complexity of Disease Progression: Disease progression is influenced by various factors beyond the linear model considered in the simulation. Incorporating a more comprehensive understanding of disease progression could enhance the accuracy of the simulation's predictions.
  6. Limited Discussion of Practical Implications: While the simulation provides insights into the interaction between BoNT therapy and disease progression, the article could benefit from a more in-depth discussion of how these findings might impact clinical practice, patient management, and future research directions.

In conclusion, the article presents a novel simulation-based exploration of the complex relationship between BoNT therapy, disease progression, and treatment outcomes. While the study offers valuable insights, its simplified model and lack of validation against real-world data are limitations that should be considered when interpreting its implications for clinical practice.

Please correct the syntax and grammatical errors. There are many dangling modifiers . 

Author Response

Strengths of the Article:

  1.  Novel Approach: The article employs a simulation-based approach to investigate the complex interplay between the effects of Botulinum neurotoxin (BoNT) therapy and disease progression. This approach provides a fresh perspective on understanding long-term treatment outcomes.
  2. Clinical Relevance: The study addresses an important clinical concern – the occurrence of secondary treatment failures in patients undergoing BoNT therapy. By modeling how the response to treatment and disease progression interact, the study sheds light on potential reasons for variations in treatment outcomes.
  3. Patient Self-Assessment: The use of patient self-assessment, including a 21-point Lickert-scale, adds credibility to the study's findings. Patient-reported outcomes are a valuable source of information for assessing treatment efficacy.
  4. Real-world Application: The study acknowledges the clinical practice of starting the next BoNT injection cycle before the full effect of the previous injection has diminished. This real-world context makes the simulation results more applicable to clinical decision-making.

Limitations of the Article:

  1. Simplified Model: The simulation employs a linear model to represent disease progression and BoNT therapy effects. While this simplification facilitates mathematical analysis, real-world disease progression is likely more complex and nonlinear.

  2. Lack of Validation: The simulation results are not validated against actual patient data or clinical trials. While the approach is insightful, validation against real-world outcomes would strengthen the conclusions.
  3. Single Disease Focus: The study primarily focuses on cervical dystonia, limiting the generalizability of its findings to other disease entities treated with BoNT.
  4. Lack of Statistical Analysis: The article does not utilize statistical analysis to support its conclusions, potentially leaving room for interpretation bias.
  5. Complexity of Disease Progression: Disease progression is influenced by various factors beyond the linear model considered in the simulation. Incorporating a more comprehensive understanding of disease progression could enhance the accuracy of the simulation's predictions.
  6. Limited Discussion of Practical Implications: While the simulation provides insights into the interaction between BoNT therapy and disease progression, the article could benefit from a more in-depth discussion of how these findings might impact clinical practice, patient management, and future research directions.

In conclusion, the article presents a novel simulation-based exploration of the complex relationship between BoNT therapy, disease progression, and treatment outcomes. While the study offers valuable insights, its simplified model and lack of validation against real-world data are limitations that should be considered when interpreting its implications for clinical practice.

Comments on the Quality of English LanguagePlease correct the syntax and grammatical errors. There are many dangling modifiers . 

We are completely in line with reviewer´s 3 arguments and have added a “strengths and limitations” section.

It has to be kept in mind that the simulation of the BoNT therapy is a non-linear iterative procedure which yields a non-linear (declining exponential) outcome curve. Only the progression is simulated as a linear function and the interaction with BoNT therapy as a summation effect. This has been done since little is known about the progression. But in principle this approach can be made much more complex. This is mentioned now.

Comparison of the model with patient data will be presented in a subsequent paper.

Reviewer 3 is absolutely right: Our model is not restricted to cervical dystonia and BoNT therapy.

Of course, it can be used for other disease entities and other treatments. This is now mentioned more clearly.

Statistical analysis will come into play when results of the model and patients´ data ere compared.

Reviewer 3 is right: According to the model disease progression is of relevance for many patients. But little is known about disease progression. Therefore this point is picked-up and included in the limitations section. Further studies are recommended to know more about disease progression.

Reviewer 3 addresses a highly relevant aspect: the implications for clinical practice. Our plan is to present the mathematical procedure in the present pilot paper and then to compare it with real-world data in a second paper. Having demonstrated the value of our approach a better basis for this necessary discussion is provided.

To our opinion reviewer 3 has picked-up the concept and the strengths and limitations of our approach very clearly so that he in principle designs the next steps for validation and interpretation. His thoughts are in line with our plan for subsequent papers.

We are very thankful for the suggestions, especially for the suggestion to add “a strengths and limitations section”.   

Reviewer 4 Report

A clear rationale for the choice of method for mathematical modeling of therapeutic outcomes discussed in the manuscript is required. Why was the number of patients used for the study sufficient?

The reference list needs to be expanded. It also contains more than half of the publications older than 10 years, which needs to be revised.

Author Response

A clear rationale for the choice of method for mathematical modeling of therapeutic outcomes discussed in the manuscript is required.

Why was the number of patients used for the study sufficient?

The reference list needs to be expanded. It also contains more than half of the publications older than 10 years, which needs to be revised.

We are thankful for this helpful comment of reviewer 4. The rationale for the choice of method is mentioned now and discussed in more detail.

The present paper only presents the simulation method. In a subsequent paper this method will be compared to patients´ data. Then the number of patients will become more relevant.

We have expanded the references and have included more recent publications.

Reviewer 5 Report

Very well designed work

Although not my area of expertise, secondary treatment failure represent a challenge in this setting

This statement should be better supported by results:

"We therefore think that for several disease entities in many centers a dose adaptation is per formed during long-term BoNT therapy compensating progression of the underlying disease entity" - line 257

Line 308: EXEL -> EXCEL

Author Response

Very well designed work

Although not my area of expertise, secondary treatment failure represent a challenge in this setting

This statement should be better supported by results:

"We therefore think that for several disease entities in many centers a dose adaptation is per formed during long-term BoNT therapy compensating progression of the underlying disease entity" - line 257

Line 308: EXEL -> EXCEL

Reviewer 5 addresses an important point:

To support our opinion and this statement a multi-center retrospective study has to be performed analysing the relation between duration of treatment and dose per session for different disease entities. This problem is now addressed in the revised manuscript.

Is corrected now.

Reviewer 6 Report

This is an interesting manuscript.

My comments are as follows, mainly about the old citations used......

Citations 1-3 are very old and should be replaced with more recent publications

Citations 4&5 should be replaced by reference to the US Prescribing Information for licensed BoNT products, which contain information on the nanograms of toxin used for clinical treatments

Line 39                       What does “The traumatic mode of application” mean?

Citations 12 & 13 are very old and should be replaced with more recent publications

Lines 48-49               There are well-established, modern definitions of STF.  New definitions should not be proposed.

Walter, U., Muhlenhoff, C., Benecke, R., Dressler, D., Mix, E., Alt, J., Wittstock, M., Dudesek, A., Storch, A., & Kamm, C. (2020). Frequency and risk factors of antibody-induced secondary failure of botulinum neurotoxin therapy. Neurology, 94(20), e2109-e2120. https://doi.org/10.1212/WNL.0000000000009444

Lines 59-60               What does “ progredient” mean?

Citations 24, 25, 26 and 27 are very old and should be replaced with more modern publications.  In particular, data from treatment of patients nearly 25 years ago are in the earlier days of understanding how to use BoNT successfully.  More modern data on therapies should be cited.

Line 84                       “my” should be “ why”

Citations 30 & 31 are incomplete

All equations should be numbered.  They should not be repeated – for example, lines 126 and 289 repeat

Line 308                     “EXEL” should be “EXCEL”

The legend to Figure 2 repeats what is included in the main text and should be changed

What are Figures 3C and 3D showing?  No reference is made to them in the main text before the legend in Figure 4

Citation 36 is incomplete

Minor improvements needed

Author Response

This is an interesting manuscript.

My comments are as follows, mainly about the old citations used......

Citations 1-3 are very old and should be replaced with more recent publications

Citations 4&5 should be replaced by reference to the US Prescribing Information for licensed BoNT products, which contain information on the nanograms of toxin used for clinical treatments

Line 39                       What does “The traumatic mode of application” mean?

Citations 12 & 13 are very old and should be replaced with more recent publications

Lines 48-49               There are well-established, modern definitions of STF.  New definitions should not be proposed.

Walter, U., Muhlenhoff, C., Benecke, R., Dressler, D., Mix, E., Alt, J., Wittstock, M., Dudesek, A., Storch, A., & Kamm, C. (2020). Frequency and risk factors of antibody-induced secondary failure of botulinum neurotoxin therapy. Neurology, 94(20), e2109-e2120. https://doi.org/10.1212/WNL.0000000000009444

Lines 59-60               What does “ progredient” mean?

Citations 24, 25, 26 and 27 are very old and should be replaced with more modern publications.  In particular, data from treatment ofpatients nearly 25 years ago are in the earlier days of understanding how to use BoNT successfully.  More modern data on therapies should be cited.

Line 84                       “my” should be “ why”

Citations 30 & 31 are incomplete

All equations should be numbered.  They should not be repeated – for example, lines 126 and 289 repeat

Line 308                     “EXEL” should be “EXCEL”

The legend to Figure 2 repeats what is included in the main text and should be changed

What are Figures 3C and 3D showing?  No reference is made to them in the main text before the legend in Figure 4

Citation 36 is incomplete

We respect previous work. That is the reason why we mention old literature.

Nevetheless, we have added more recent publications.

We have tried to get the prescribing information.

This is explained now.

We add more recent references.

We do not propose a new definition for STF. But we distinguish between STF in the strict sense and insufficient treatment.

This is explained now.

Recent data are added.

This is corrected.

This is corrected.

Equations are not repeated now.

This is corrected.

We have modified and expanded the legends to Fig. 2 to 4. We are sorry for the confusion, resulting from up-loading an old version of Fig. 3. The new version is now consistent with the extended legend and the main text.

This is corrected.

Round 2

Reviewer 3 Report

Thank you for revising the manuscript.

1. Some of the references (Fabbri et al.)should be replaced by  more recent systematic review and meta analysis : Rahman E, Alhitmi HK, Mosahebi A. Immunogenicity to Botulinum Toxin Type A: A Systematic Review With Meta-Analysis Across Therapeutic Indications. Aesthet Surg J. 2022 Jan 1;42(1):106-120. doi: 10.1093/asj/sjab058. PMID: 33528495.

Reviewer 4 Report

Technical corrections of text are required.

I meant that the manuscript needs to be checked for punctuation and text formatting. For example, remove an extra period (line 42); non-standard paragraph spacing (lines 49-80 and onward in the text); the bibliography needs a consistency check.

Author Response

I meant that the manuscript needs to be checked for punctuation and text formatting. For example, remove an extra period (line 42); non-standard paragraph spacing (lines 49-80 and onward in the text); the bibliography needs a consistency check.

Thank you for your thoughtful review and for pointing out the issues related to punctuation and text formatting. Our native speaker has worked on linguistic fine-tuning and made the necessary corrections. This includes removing the extra period in line 42 and standardizing paragraph spacing throughout the document. Additionally, we have ensured that the bibliography now follows a consistent format. We have accepted all the revisions for ease of formatting and text editing. For more details, please refer to the previous version.